# The Covid-19 pandemic and mental health of first-year college students: Examining the effect of Covid-19 stressors using longitudinal data

**Jane Cooley Fruehwirth**[1]*, **Siddhartha Biswas**[1], **Krista M. Perreira**[2]

**1** Department of Economics, University of North Carolina at Chapel Hill, Chapel Hill, North Carolina, United States of America, **2** Department of Social Medicine, School of Medicine, University of North Carolina at Chapel Hill, Chapel Hill, North Carolina, United States of America

\* jane_fruehwirth@unc.edu

## Abstract

### Purpose

The Covid-19 pandemic has brought unprecedented stress to students and educational institutions across the world. We aimed to estimate the effect of the pandemic on the mental health of college students.

### Methods

We used data on 419 first-year students (ages 18–20) at a large public university in North Carolina both before (October 2019-February 2020) and after (June/July 2020) the start of the Covid-19 pandemic. After evaluating descriptive data on mental health and stressors by students' demographic characteristics, we estimated the associations between Covid-19 stressors (including work reductions, health, distanced learning difficulties and social isolation) and mental health symptoms and severity controlling for students' pre-pandemic mental health, psychosocial resources, and demographic characteristics.

### Results

We found that the prevalence of moderate-severe anxiety increased from 18.1% before the pandemic to 25.3% within four months after the pandemic began; and the prevalence of moderate-severe depression increased from 21.5% to 31.7%. White, female and sexual/ gender minority (SGM) students were at highest risk of increases in anxiety symptoms. Non-Hispanic (NH) Black, female, and SGM students were at highest risk of increases in depression symptoms. General difficulties associated with distanced learning and social isolation contributed to the increases in both depression and anxiety symptoms. However, work reductions as well as Covid-19 diagnosis and hospitalization of oneself, family members or friends were not associated with increases in depression or anxiety symptoms.

**Data Availability Statement:** The UNC ethics committee has deemed that the data contain potentially sensitive information and that there is a possibility of deductive disclosure, so that the

human subjects approval does not allow us to share the de-identified data. The de-identified data will be made available upon request to the Deputy Director of Research at the UNC Carolina Population Center (TransitionsDataRequest@office.unc.edu) with an appropriate restricted use data agreement in place.

**Funding:** This research was supported by the Carolina Population Center and its National Institutes of Health (NIH)/National Institute of Child Health and Human Development (NICHD) Grant Award Number P2C HD50924 (JF), the Integrating Special Populations/ North Carolina Translational and Clinical Sciences Institute through Grant Award Number ILITR002489 (KP). We also thank the Economics department and Office of Undergraduate Research at UNC-Chapel Hill for funding (JF). The content is solely the responsibility of the authors and does not necessarily represent the official views of the NIH or the funders. The funders had no role in study design, data collection and analysis, decision to publish, or preparation of the manuscript.

**Competing interests:** The authors have declared that no competing interests exist.

## Conclusion

Colleges may be able to reduce the mental health consequences of Covid-19 by investing in resources to reduce difficulties with distance learning and reduce social isolation during the pandemic.

## Introduction

The Covid-19 pandemic has brought unprecedented stress to the educational system in the US, not least to colleges and their students. Colleges have faced the difficult decision of whether to reopen in the 2020/21 academic year and risk becoming a super spreader of the virus or to take classes online in face of potential losses of revenue and diminished ability to support the most vulnerable students. Many colleges are facing significant financial stress regardless of the chosen path [1].

The pandemic has also brought unprecedented stress to college students, starting with the transition to online instruction over spring break 2020 at many universities [2]. This is further exacerbated by the long summer of social isolation from the pandemic for many, lost employment, and uncertainty about the structure of courses and living arrangements in the 2020/21 academic year [3, 4]. Exploiting data collected for the same students pre- and during the pandemic, we provide new evidence on the effects of the pandemic on mental health of first-year college students, focusing particularly on the effects of different Covid-stressors.

Even prior to these events, universities nationwide were struggling with a growing mental health crisis on their campuses. Research finds that young adults 18–25 in the US experienced large increases (63 percent over the past decade) in major depressive episodes [5]. In a national sample of universities, the rate of mental health treatment increased from 19% to 34% between 2007 and 2017 [6]. Among students seeking mental health treatment on campuses, anxiety and depression were the most frequent concerns [7]. The trends for adolescents are particularly troubling given the far-reaching impact of mental illness on physical health, educational outcomes, and employment outcomes well into adulthood [8–11]. This study focuses on a diverse sample of first-year students. The first year is understood to be a particularly challenging year for students given the transition to a new school environment and the increased independence students experience [12, 13]. We surveyed first-year students enrolled in a large public university in North Carolina both before (October 2019-February 2020) and after (June/July 2020) the start of the Covid-19 pandemic. We estimate an overall effect of the pandemic by comparing changes in anxiety/depression for the same student from pre- to during the pandemic. Because of multiple rounds of data collection prior to the pandemic, we are able to test that changes in anxiety/depression are not driven by pre-existing trends in anxiety/depression over the first year of college. We exploit rich data on Covid-19-related stressors (e.g., work reductions by either students or their parents, Covid-19 diagnosis or hospitalization of oneself, family members, or friends, distanced learning, and social isolation) to establish the extent to which changes in anxiety/depression symptoms were predicted by pandemic-related factors. These stressors are motivated by a prior literature showing the importance of financial stress [14], academic stress [15], and social isolation for mental health [16].

The main contribution of our study is to provide early estimates of the effect of the Covid-19 pandemic on anxiety/depression symptoms of US college students in their first year at university. Other studies have also estimated how mental health symptoms for college students have changed from pre- to during the pandemic. For instance, a repeated cross-section study of US college students compared depression and anxiety rates in Fall 2019 for 58 campuses to late March-May

2020 for 16 campuses and found small increases in depression from 35.7 to 40.9% and no changes in anxiety [17]. Another cross-sectional study found that 19% of college students in Fall 2020 reported that their emotional health was far worse since the pandemic began [18].

A growing number of cross-sectional studies raise concerns about the effect of the pandemic on college student mental health [17–26], and some speak to potential factors that are related to our Covid-19 related stressors. One study found that depression and anxiety rates for undergraduates in 9 US public research universities in May/July 2020 were higher for those who had trouble adapting to distanced learning [25]. Another conducted at a public university in the US in April 2020 found that worse mental health was associated with employment losses, difficulties focusing on academic work and concern about Covid-19 [24]. Two studies of college students in China using post-pandemic data found that family income stability was negatively associated with anxiety symptoms and that Covid-19 diagnosis of family or friends was positively associated with anxiety and depression symptoms [27, 28]. A study of college students in Turkey found that students were more anxious about the effects of Covid-19 on relatives than on themselves [29]. Another study based on a sample of young adults in India found significant associations between mental health and economic stressors [30].

Our study extends the previous research by focusing on first-year students, a particularly vulnerable population, examining the effects of a broad set of Covid-related stressors, and using longitudinal data. Using longitudinal data is an important extension for several reasons. First, it directly addresses differential selection into survey participation during the pandemic compared to pre-pandemic, which would affect the internal validity of repeated cross-section designs. It also addresses concerns about imperfect recall that may exist in cross-sectional designs where respondents are asked to compare current mental health to a previous point in time. Second, it permits us to investigate underlying causes after accounting for key confounds, namely pre-existing mental health and psychosocial resources.

We are only aware of two longitudinal surveys of college student mental health. One compares anxiety and depression symptom severity in April 2020 for 205 college students at a large public university and found significant increases in severity compared to 2 to 8 months earlier [31]. They found that cognitive and behavioral avoidance, online social engagement and problematic internet use were predictors of these changes. Another compares wellness behaviors at the beginning to the end of Spring semester 2020 for first-year students and finds modest effects of the pandemic [32]. An important contribution relative to these studies is that we consider the effects of a different set of determinants, namely job loss, changes in social isolation, challenges with distanced learning and health. These stressors help inform the difficult decisions universities have faced about whether to invite students back to campus or to take classes online and additional support students may need during the pandemic.

We hypothesize that among first-year college students both anxiety and depression symptoms will increase after the onset of the Covid-19 pandemic. In addition, the magnitude of changes in anxiety and depression symptoms will vary by race/ethnicity, female/male sex, sexual/gender minority (SGM) identity, and first-generation college (FGC) student status. Pandemic-related stressors such as work reductions, distance learning, Covid-19 diagnoses and hospitalizations, and social isolation will also vary across these student populations and will be associated with increased anxiety and depression symptoms.

## Methods

### Data

This study was approved by the University of North Carolina-Chapel Hill's Institutional Review Board (reference 19–1947). Survey data were collected via two 25-minute Qualtrics

surveys completed on-line as part of Waves I and II of the Transitions Study. Consent was obtained by virtue of agreeing to participate in the on-line Qualtrics survey and data were analyzed anonymously. Wave I was initiated in October/November 2019 with an email invitation to a random sample of in-state, first-year college students age 18 or older and enrolled in the selected public university. In January/February 2020, we expanded the sample to include all enrolled first-year students. Participants who did not respond to the initial email invitation were sent a follow-up invitation offering a $10 gift card to participants. In June/July 2020, roughly four months after the start of the pandemic, we invited 738 of our Wave I respondents who indicated a willingness to participate in additional surveys to complete a follow-up survey and offered participants a $15 gift card. Consistent with many online surveys [33], our Wave I response rate was 32% (N = 1124). Our Wave II response rate was 64 percent (N = 472). Our analytic sample for this study includes 419 participants who completed both the Wave I and II surveys and who have no missing data on mental health measures or Covid-19 stressors.

## Setting

Data for this study were collected at a large public university in NC. In NC, the Governor issued a stay-at-home order in late March. At about the same time, the university made the decision to send most students home and moved classes online for the remaining five weeks of the semester and summer sessions. Until mid-May, confirmed Covid-19 cases in NC were initially below an average of 500 per day [34]. After the stay-at-home order was lifted in mid-May, average cases per day in NC rose to 2000 [34].

## Measures

**Mental health.** We measured depression and anxiety symptoms at both Waves I and II. To measure depression, we used the Patient Health Questionnaire Depression Scale (PHQ-8), a measure of eight depression symptoms occurring "not at all" (0) to "nearly every day" (3) over the past two weeks [35]. To measure anxiety symptoms, we used the Generalized Anxiety Disorder scale (GAD-7), a measure of seven anxiety symptoms occurring "not at all" (0) to "nearly every day" (3) over the past two weeks [36]. For both measures, we summed across responses to create a continuous measure of symptoms and also created a dichotomous measure of moderate-severe symptoms for scores of 10 or more [35, 36]. To ease interpretation in our regression analyses, measures of anxiety and depression symptoms were standardized to have a mean of zero and standard deviation of one. In our sample, Cronbach's alphas were .90 and .88 for the GAD-7 and PHQ-8, respectively.

**Covid-19 stressors.** First, we measure two economic stressors–student and parent work reductions. Students were first asked whether they were employed at Wave I. Then, at Wave II, students were asked whether they or their parents had lost a paid job, were furloughed, or had their hours reduced. Second, we measured educational stressors by asking students to rate the difficulty of engaging in nine activities on a 4-point Likert scale. An exploratory factor analysis of their responses identified two factors–distance learning and educational technology. For each factor, we utilized standardized factor scores with means of zero and standard deviation of one. The higher factor scores for distance learning indicate greater difficulties with finding support needed for courses (e.g. tutoring and office hours), accessing the learning materials needed, adapting to the distanced learning format, finding a quiet space to work, and making time for course work. Higher factor scores for education technology indicate greater difficulties with accessing the internet and obtaining the technology (e.g., computers and software) needed for distance learning. Third, we measured Covid-19 health stressors. Covid-19 diagnosis and hospitalization identified, respectively, whether students, their family members, or

their friends had been diagnosed with Covid-19 or hospitalized with Covid-19. Finally, at both Waves I and II, we measured whether a student felt isolated from others either always/usually or rarely/never [37].

**Psychosocial resources.** At Wave I, we included three measures to identify students' psychosocial resources. We measured resilience averaging responses to six items (accounting for reverse-coding) measured on a 5-point Likert scale using the Brief Resilience Scale (BRS).[38] The measure was standardized to have a mean of zero and standard deviation of one in the full Wave I sample. Similarly, we measured coping using the 4-item Brief Resilient Coping Scale (BRCS) [39]. Based on the sum of these items, we defined three categories–low-resilient copers with scores less than 13, medium-resilient copers with scores of 13–17, and high-resilient copers with scores greater than 17. Using the Multidimensional Scale of Perceived Social Support (MSPSS), we measured perceived social support in three domains–family, friend and significant other [40]. For each domain, we averaged four domain-specific questions whose responses ranged from 1 (strongly disagree) to 5 (strongly agree). Measures of psychosocial resources had Cronbach's alphas ranging from .60 for the BRCS to .85 for the BRS and .89-.93 for each domain of the MSPSS.

**Demographic characteristics.** Wave I data include key demographics–race/ethnicity, male/female sex, sexual orientation and gender identity, and whether the student received free or reduced-price lunch in high school. Free or reduced-price lunch status provides a rough proxy for low-income. We classified students as Hispanic if they report Hispanic ethnicity regardless of race, non-Hispanic (NH) Black, NH White, NH Asian and NH Other for any other race/ethnicity, including mixed-race students. We defined a sexual or gender minority (SGM) student as a student who reported any sexual orientation other than heterosexual, a transgender identity, or a gender identity other than their sex at birth. We defined a first-generation college (FGC) student to be one for whom neither parent had completed a 4-year post-secondary degree.

## Analysis

In this study, we first evaluated mean differences in the characteristics of participants at Wave I (pre-pandemic) and participants who also completed Wave II (four months into the pandemic). Second, we investigated whether an upward trend in anxiety and depression symptoms existed pre-pandemic by examining whether symptoms were significantly higher in January/February compared to October/November. Third, we examined differences in Covid-19 stressors by demographic groups. Finally, we estimated the associations between Covid-19 stressors and moderate-severe symptoms using logistic regressions. These models control for mental health, social isolation, psychosocial resources, and demographic characteristics at Wave I. Models also include an indicator variable for the week of the follow-up survey. In additional analyses, we estimated models separately by those with and those without anxiety and depression symptoms at Wave I and dropped controls for Wave I symptoms. In supplemental analyses, we compare these results to continuous models of anxiety/depression symptom severity using ordinary least squares regressions.

## Results

### Sample characteristics

Our longitudinal sample is roughly representative of first-year students at the university and the pandemic did not appear to systematically affect the participation of students in Wave II (Table 1). In our longitudinal sample of students who responded to both survey waves, 61.6% were NH White, 6.7% were NH Black, 18.1% were NH Asian, 8.4% were Hispanic of any race. This is roughly comparable to the reported demographics of the university's first-year student

**Table 1. Comparisons of means between the Transitions Study cross-sectional (Wave I) and longitudinal (Wave I—Wave II) samples.**

| | Cross-Sectional Sample | | Longitudinal Sample | |
|---|---|---|---|---|
| | Mean[b] | (s.e.) | Mean[b] | (s.e.) |
| Demographic characteristics | | | | |
| NH White[a] | 0.627 | (0.016) | 0.616 | (0.024) |
| NH Black[a] | 0.068 | (0.008) | 0.067 | (0.012) |
| Hispanic | 0.087 | (0.009) | 0.084 | (0.014) |
| NH Asian[a] | 0.157 | (0.012) | 0.181 | (0.019) |
| NH Other[a] | 0.060 | (0.008) | 0.053 | (0.011) |
| Female | 0.664 | (0.015) | 0.704 | (0.022) |
| SGM[a] | 0.168 | (0.012) | 0.181 | (0.019) |
| FGCS[a] | 0.169 | (0.012) | 0.174 | (0.019) |
| Age | 18.926 | (0.013) | 18.909 | (0.019) |
| Free/reduced price lunch | 0.153 | (0.012) | 0.152 | (0.018) |
| Mental health (pre-pandemic) | | | | |
| Anxiety symptoms (GAD-7) | 5.190 | (0.155) | 5.413 | (0.224) |
| Moderate to severe anxiety | 0.178 | (0.012) | 0.181 | (0.019) |
| Depression symptoms (PHQ-8) | 6.109 | (0.160) | 6.229 | (0.236) |
| Moderate to severe depression | 0.204 | (0.013) | 0.215 | (0.020) |
| Psychological resources and social support (pre-pandemic) | | | | |
| Social isolation | 0.191 | (0.013) | 0.219 | (0.021) |
| Brief resilience scale | 3.383 | (0.024) | 3.354 | (0.035) |
| Perceived social support, friends | 4.117 | (0.025) | 4.130 | (0.039) |
| Perceived social support, family | 4.048 | (0.029) | 4.029 | (0.045) |
| Perceived social support, significant other | 3.920 | (0.032) | 3.978 | (0.048) |
| Brief resilient coping, low | 0.245 | (0.014) | 0.237 | (0.021) |
| Brief resilient coping, moderate | 0.525 | (0.016) | 0.513 | (0.025) |
| Brief resilient coping, high | 0.229 | (0.014) | 0.249 | (0.021) |
| N | 966 | 966 | 419 | 419 |

[a]Abbreviations: Sexual/Gender Minority, SGM; First-Generation College Student, FGCS; Non-Hispanic, NH; standard errors, s.e.

[b]Note: We found no statistically significant differences between means for the cross-sectional and longitudinal samples.

population in 2019/20–55.7% NH White, 8.9% NH Black, 12.3% NH Asian, 9% Hispanic of any race.[41] Furthermore, 17.4% of our respondents are FGC students, slightly lower than the percentage (18.9%) of FGC students at the university [41]. Comparing the means of characteristics of our longitudinal sample (Wave I and II) with the characteristics of the cross-sectional sample (Wave I only), we find no significant differences in mean characteristics.

## Changes in mental health by demographic characteristics

The prevalence of moderate-severe anxiety symptoms increased by 40 percent from 18.1% pre-pandemic to 25.3% mid-pandemic (Table 2). Additionally, changes in prevalence of moderate-severe anxiety symptoms varied by demographic group. Before the pandemic, NH Black students reported the highest prevalence of moderate-severe anxiety (32.1%) and Hispanic students reported the lowest prevalence (14.3%) of any racial/ethnic group. But prevalence rates for these two groups did not increase after the start of the pandemic. In contrast, the prevalence of moderate-severe anxiety increased significantly among NH White, female, and non-FGC students. Among SGM students, prevalence rates increased the most, growing from 28.9% pre-pandemic to 46.1% mid-pandemic.

**Table 2. Pre- and post-pandemic comparison of means of moderate-severe anxiety and depression symptoms among the Transitions Study longitudinal aample (N = 419), by demographic characteristics.**

| | Pre-pandemic (Wave I) | Post-pandemic (Wave II) | Wave II—Wave 1 |
|---|---|---|---|
| | Mean | Mean | Difference of Means |
| Moderate to Severe Anxiety | | | |
| Overall | 0.181 | 0.253 | 0.072** |
| NH White[a] | 0.178 | 0.256 | 0.078** |
| NH Black[a] | 0.321 | 0.357 | 0.036 |
| Hispanic | 0.143 | 0.171 | 0.029 |
| NH Asian[a] | 0.145 | 0.211 | 0.066 |
| NH Other[a] | 0.227 | 0.364 | 0.136 |
| Female | 0.200 | 0.281 | 0.081** |
| Male | 0.137 | 0.185 | 0.048 |
| Non-SGM[a] | 0.157 | 0.207 | 0.050* |
| SGM[a] | 0.289 | 0.461 | 0.171** |
| Non-FGCS[a] | 0.182 | 0.266 | 0.084*** |
| FGCS[a] | 0.178 | 0.192 | 0.014 |
| Moderate to Severe Depression | | | |
| Overall | 0.215 | 0.317 | 0.103*** |
| NH White[a] | 0.198 | 0.283 | 0.085** |
| NH Black[a] | 0.321 | 0.607 | 0.286** |
| Hispanic | 0.257 | 0.286 | 0.029 |
| NH Asian[a] | 0.184 | 0.276 | 0.092 |
| NH Other[a] | 0.318 | 0.545 | 0.227 |
| Female | 0.231 | 0.353 | 0.122*** |
| Male | 0.177 | 0.234 | 0.056 |
| Non-SGM[a] | 0.169 | 0.248 | 0.079** |
| SGM[a] | 0.421 | 0.632 | 0.211*** |
| Non-FGCS[a] | 0.202 | 0.315 | 0.113*** |
| FGCS[a] | 0.274 | 0.329 | 0.055 |

*** $p < 0.001$

** $p < 0.05$

* $p < 0.1$

[a]Abbreviations: Sexual/Gender Minority, SGM; First-Generation College Student, FGCS; Non-Hispanic, NH.

Similarly, the prevalence of moderate-severe depression symptoms increased by 48 percent from 21.5% to 31.7%. The prevalence of moderate-severe depression symptoms also varied by demographic group. Both NH Black and Hispanic students reported high prevalence of moderate-severe depression symptoms, 32.1% and 25.7% respectively. However, after the start of the pandemic the prevalence of moderate-severe depression only increased significantly (90 percent) for NH Black students. SGM students also reported a high prevalence of moderate-severe depression symptoms (42%) which increased significantly (50 percent) after the Covid-19 pandemic began. We found that there were no statistically significant changes in moderate-severe anxiety or depression symptoms over the first year prior to the pandemic (Table 3).

## Covid-19 related stressors by demographic characteristics

Demographic variations in Covid-19 stressors may partially explain demographic differences in mental health (**Table 4**). We found that a high percentage of formerly employed students

**Table 3. Test for trends in anxiety and depression symptoms prior to the pandemic among in-state students of the Transitions Study cross-sectional sample (N = 807).**

| | Moderate to Severe Anxiety | | Moderate to Severe Depression | |
|---|---|---|---|---|
| | Difference of Means (Jan/Feb '20—Oct/Nov '19) | *p*-value | Difference of Means (Jan/Feb '20—Oct/Nov '19) | *p*-value |
| Overall | 0.010 | 0.724 | 0.049 | 0.114 |
| NH White[a] | -0.010 | 0.791 | 0.028 | 0.459 |
| NH Black[a] | 0.083 | 0.557 | 0.075 | 0.630 |
| Hispanic | -0.027 | 0.788 | 0.109 | 0.321 |
| NH Asian[a] | 0.055 | 0.435 | 0.042 | 0.571 |
| NH Other[a] | 0.143 | 0.289 | 0.246 | 0.085* |
| Female | 0.014 | 0.721 | 0.032 | 0.415 |
| Male | -0.003 | 0.950 | 0.074 | 0.121 |
| Non-SGM[a] | -0.006 | 0.846 | 0.039 | 0.205 |
| SGM[a] | 0.084 | 0.330 | 0.072 | 0.445 |
| Non-FGCS[a] | 0.002 | 0.947 | 0.040 | 0.224 |
| FGCS[a] | 0.054 | 0.421 | 0.085 | 0.290 |

[a]Abbreviations: Sexual/Gender Minority, SGM; First-Generation College Student, FGCS; Non-Hispanic, NH.

*** $p < 0.001$

** $p < 0.05$

* $p < 0.1$

experienced work reductions (58.6%), had parents who experience work reductions (36%), knew someone who had been diagnosed (30%) or hospitalized with Covid-19 (10.3%), and reported feeling socially isolated either before (21.9%) or during the pandemic (30%).

These stressors differed significantly by demographic characteristics. Hispanic students (compared to NH White students) and FGC students (compared to non-FGC students) more frequently reported that their parents had experienced a work reduction. Hispanic students (compared to NH White students), females (compared to males), SGM (compared to non-SGM) students, and FGC (compared to non-FGC) students reported significantly more difficulties with distance learning. No significant differences in access to educational technology were reported by demographic group. Considering themselves, family, and friends, SGM students also reported the highest rates of Covid-19 diagnosis and Hispanic students reported the highest rates of Covid-19 hospitalizations. Finally, SGM students reported significantly greater social isolation than non-SGM students after the pandemic started. However, there were no significant differences in social isolation between these two groups pre-pandemic. In contrast, pre-pandemic social isolation was significantly higher for FGC students than for non-FGC students. Yet four months into the pandemic, there were no differences in social isolation between these groups.

## Effect of Covid-19 stressors on mental health

We turn now to estimating associations between Covid-19 related stressors and our two mental health outcomes–moderate-severe anxiety and depression symptoms–while controlling for a rich set of pre-pandemic characteristics (Tables 5 and 6). We report odds ratios and marginal effects estimates (i.e., the change in probability of the outcome for a small or discrete change in an explanatory variable) for the regressions on the overall sample. We only report marginal effects in the remaining regressions. The remaining regressions in Table 5 (6) were estimated separately for those with and those without Wave I anxiety (depression) symptoms and did

**Table 4. Means of Covid-19 stressors among the Transitions Study longitudinal sample, by demographic characteristic (N = 419).**

| By race | Overall | NH White[a] | NH Black[a] | Hispanic | NH Asian[a] | NH Other[a] |
|---|---|---|---|---|---|---|
| Employed (Wave I) | 0.432 | 0.442 | 0.464 | 0.629** | 0.316** | 0.364 |
| Work reduction (student) | 0.253 | 0.260 | 0.321 | 0.343 | 0.184 | 0.182 |
| Work reduction (parent) | 0.358 | 0.353 | 0.321 | 0.571** | 0.316 | 0.273 |
| Distanced learning | 0.000 | -0.022 | 0.119 | 0.308* | -0.131 | 0.070 |
| Education technology | 0.000 | -0.062 | -0.053 | 0.135 | 0.173* | -0.019 |
| Covid-19 diagnosis | 0.296 | 0.310 | 0.357 | 0.457* | 0.145*** | 0.318 |
| Covid-19 hospitalization | 0.103 | 0.093 | 0.179 | 0.229** | 0.066 | 0.045 |
| Social isolation (Wave II) | 0.284 | 0.279 | 0.393 | 0.171 | 0.289 | 0.364 |
| Social isolation (Wave I) | 0.219 | 0.215 | 0.296 | 0.242 | 0.187 | 0.238 |
| By other characteristics | Female | Male | Non-SGM[a] | SGM[a] | Non-FGCS[a] | FGCS[a] |
| Employed (Wave I) | 0.464 | 0.355** | 0.431 | 0.434 | 0.436 | 0.411 |
| Work reduction (student) | 0.281 | 0.185** | 0.239 | 0.316 | 0.260 | 0.219 |
| Work reduction (parent) | 0.393 | 0.274** | 0.344 | 0.421 | 0.329 | 0.493*** |
| Distanced learning | 0.084 | -0.201*** | -0.081 | 0.363*** | -0.076 | 0.361*** |
| Education technology | -0.015 | 0.036 | 0.000 | 0.000 | -0.016 | 0.077 |
| Covid-19 diagnosis | 0.292 | 0.306 | 0.274 | 0.395** | 0.283 | 0.356 |
| Covid-19 hospitalization | 0.102 | 0.105 | 0.102 | 0.105 | 0.092 | 0.151 |
| Social isolation (Wave II) | 0.298 | 0.250 | 0.251 | 0.434*** | 0.286 | 0.274 |
| Social isolation (Wave I) | 0.230 | 0.193 | 0.215 | 0.239 | 0.192 | 0.353*** |

[a]Abbreviations: Sexual/Gender Minority, SGM; First-Generation College Student, FGCS; Non-Hispanic, NH.

Note

*** $p < 0.001$

** $p < 0.05$

* $p < 0.1$ are p-values from tests for whether means are statistically different from the respective demographic control group: White, Female, Non-SGM, and Non-FGCS.

not control for Wave I anxiety (depression) symptoms as a result. Odds ratios cannot be compared across model specifications when the sample or conditioning set changes, whereas marginal effects can be compared [42, 43].

We identify four main results from our regression analyses. First, neither work reductions among formerly employed students nor parent work reductions were associated with significant increases in moderate-severe anxiety or depression symptoms. The exception was for those with moderate-severe anxiety symptoms prior to the pandemic. For these students, retaining their jobs had a protective effect, whereas experiencing work reductions was associated with an increase in anxiety.

Second, students who experienced difficulties with distance learning experienced higher rates of moderate-severe anxiety and depression. A one-standard-deviation increase in distance learning challenges was associated with an 8.1 percentage point increase in moderate-severe anxiety and a 7.0 percentage point increase in moderate-severe depression. Results were similar for those with no anxiety or depression symptoms in Wave I. Marginal effects were almost 3 times higher for those with anxiety symptoms prior to the pandemic. In contrast, difficulties accessing education technology had no significant association with anxiety and depression symptoms, overall or by pre-Covid symptoms.

Third, we found no evidence that Covid-19 diagnoses or hospitalizations for oneself, family, or friends were associated with significant increases in anxiety or depression symptoms.

**Table 5. Logistic regression estimates for moderate-severe anxiety symptoms among the Transitions Study longitudinal sample (N = 419).**

| | Moderate-Severe Anxiety | | | |
| --- | --- | --- | --- | --- |
| | Odds Ratio (s.e.)[a] | Marginal Effects (s.e.)[a] | Marginal Effects (s.e.)[a] | Marginal Effects (s.e.)[a] |
| Employed (Wave I) | 0.426* | -0.108* | -0.045 | -0.905*** |
| | 0.204 | 0.061 | 0.046 | 0.345 |
| x Work reduction (student) | 1.590 | 0.059 | 0.007 | 1.470*** |
| | 0.822 | 0.065 | 0.053 | 0.501 |
| Work reduction (parent) | 1.147 | 0.017 | -0.027 | -0.260* |
| | 0.362 | 0.040 | 0.048 | 0.133 |
| Distanced learning | 1.891*** | 0.081*** | 0.082*** | 0.208*** |
| | 0.325 | 0.021 | 0.022 | 0.079 |
| Education technology | 0.918 | -0.011 | -0.015 | -0.117 |
| | 0.126 | 0.017 | 0.017 | 0.074 |
| Covid-19 diagnosis | 0.987 | -0.002 | -0.021 | 0.323* |
| | 0.370 | 0.047 | 0.051 | 0.191 |
| Covid-19 hospitalization | 1.223 | 0.026 | 0.012 | 0.421* |
| | 0.647 | 0.067 | 0.072 | 0.249 |
| Social isolation (Wave II) | 3.565*** | 0.161*** | 0.144*** | 0.763*** |
| | 1.199 | 0.039 | 0.041 | 0.208 |
| Anxiety symptoms (Wave I) | 3.576*** | 0.161*** | | |
| | 1.270 | 0.042 | | |
| Social isolation (Wave I) | 0.275*** | -0.163*** | -0.155** | -0.706*** |
| | 0.123 | 0.053 | 0.068 | 0.247 |
| $R^2$ | 0.297 | 0.297 | 0.265 | 0.570 |
| Whole sample | Yes | Yes | No | No |
| Moderate to severe anxiety symptoms in Wave I? | | | No | Yes |
| Joint significance of Covid-19 stressors | | | | |
| $p$-value | 0.002 | 0.002 | 0.011 | 0.015 |

[a]Abbreviations: standard errors, s.e.

Note: All models control for students' psychological resources and social support measures listed in Table 1, missing indicators for psychological resources and social support measures, race, gender, Sexual/gender minority identity, first-generation college student status, age, free/reduced priced-lunch, the week in which the student responded to Wave II, and a constant. Columns 3,4,7,8 show marginal effects from separate samples of students who did and did not have moderate to severe symptoms in Wave I.

*** $p < 0.001$

** $p < 0.05$

* $p < 0.1$

Fourth, social isolation significantly and profoundly influenced the risk of moderate-severe anxiety and depression even after controlling for perceptions of social isolation prior to the pandemic. We found a 16.1 percentage point increase in moderate-severe anxiety symptoms and 17.7 percentage point increase in moderate-severe depression symptoms among students who reported feeling usually or always socially isolated mid-pandemic (and had not reported social isolation pre-pandemic). Results for anxiety and depression were similar for those who did not have moderate-severe anxiety or depression symptoms in Wave I, but again markedly higher for those with moderate-severe anxiety symptoms at Wave I. Finally, we found that students who were already experiencing mental health problems pre-pandemic were at greater odds of experiencing severe symptoms mid-pandemic. In S1 Table, we show that results are similar for continuous measures of the severity of anxiety and depression symptoms.

**Table 6. Logistic regression estimates for moderate-severe depression symptoms among the Transitions Study longitudinal sample (N = 419).**

| | Moderate-Severe Depression | | | |
|---|---|---|---|---|
| | Odds Ratio (s.e.)[a] | Marginal Effects (s.e.)[a] | Marginal Effects (s.e.)[a] | Marginal Effects (s.e.)[a] |
| Employed (Wave I) | 0.589 | -0.066 | -0.037 | -0.133 |
| | 0.228 | 0.049 | 0.055 | 0.124 |
| x Work reduction (student) | 1.572 | 0.057 | 0.074 | -0.186 |
| | 0.745 | 0.059 | 0.063 | 0.153 |
| Work reduction (parent) | 1.636 | 0.062 | 0.047 | -0.006 |
| | 0.502 | 0.038 | 0.040 | 0.149 |
| Distanced learning | 1.742*** | 0.070*** | 0.077*** | 0.065 |
| | 0.300 | 0.021 | 0.025 | 0.069 |
| Education technology | 0.842 | -0.022 | -0.036* | 0.026 |
| | 0.121 | 0.018 | 0.021 | 0.062 |
| Covid-19 diagnosis | 0.974 | -0.003 | -0.044 | 0.142 |
| | 0.383 | 0.049 | 0.063 | 0.135 |
| Covid-19 hospitalization | 1.369 | 0.039 | 0.079 | -0.055 |
| | 0.760 | 0.070 | 0.077 | 0.166 |
| Social isolation (Wave II) | 4.084*** | 0.177*** | 0.214*** | 0.133 |
| | 1.357 | 0.038 | 0.037 | 0.156 |
| Depression symptoms (Wave I) | 5.240*** | 0.208*** | | |
| | 2.001 | 0.044 | | |
| Social isolation (Wave I) | 1.069 | 0.008 | 0.022 | -0.049 |
| | 0.508 | 0.060 | 0.063 | 0.178 |
| $R^2$ | 0.365 | 0.365 | 0.325 | 0.260 |
| Whole sample | Yes | Yes | No | No |
| Moderate to severe depression symptoms in Wave I? | | | No | Yes |
| Joint significance of Covid-19 stressors | | | | |
| p-value | 0.007 | 0.007 | 0.002 | 0.159 |

[a]Abbreviations: standard errors, s.e.

Note: All models control for students' psychological resources and social support measures listed in Table 1, missing indicators for psychological resources and social support measures, race, gender, Sexual/gender minority identity, first-generation college student status, age, free/reduced priced-lunch, the week in which the student responded to Wave II, and a constant. Columns 3,4,7,8 show marginal effects from separate samples of students who did and did not have moderate to severe symptoms in Wave I.

*** $p < 0.001$

** $p < 0.05$

* $p < 0.1$

## Discussion

Using longitudinal data, this study examined the effects of the Covid-19 pandemic on the mental health of first-year college students. We found that rates of moderate-severe anxiety increased 39.8 percent and rates of moderate-severe depression increased 47.9 percent from before to mid-pandemic. We also found that these changes were not driven by increasing trends in anxiety and depression symptoms resulting from typical first year stressors prior to the pandemic. With one-quarter of students experiencing moderate-severe anxiety and nearly one-third experiencing moderate-severe depression four months into the pandemic, Covid-19 will place new stress on an already stressed college system.

The difficulties associated with distance learning and the social isolation engendered by the pandemic contributed most substantially to the observed increases in anxiety and depression

symptoms among first-year college students. Hispanic, FGC, and SGM students experienced the greatest difficulties with distance learning. But neither Hispanic nor FGC students experienced significant increases in moderate-severe anxiety or depression. SGM students, on the other hand, experienced significant increases in both. Among SGM students, moderate-severe anxiety increased 59% and moderate-severe depression increased 50%. Social isolation also increased precipitously for SGM students (23.9% to 43.4%) as well as Black students (29.6% to 39.3%). Clearly, social isolation contributed to the 89% increase in depression that we observed among Black students. For Hispanics and FGC students, feelings of social isolation actually declined from 24.2% to 17.1% and 35.3% to 27.4%, respectively as these students left the university and returned to their homes. For these students, returning home may have helped to reduce the risk for mid-pandemic increases in depression and anxiety symptoms. This result is consistent with research showing lower income students, many of whom may be FGC and Hispanic, experience greater social isolation under typical university conditions [44].

Though this research provides critical insights into the effects of the Covid-19 pandemic on the mental health of first-year college students, several limitations will need to be addressed in future research. First, our results are limited to a single university and first-year students. Research on the mental health of students should be expanded to other years and to include other universities within the US. Second, our relatively small sample size prohibits us from separately evaluating the effects of stressors and psychosocial resources on mental health among the populations most at risk. Future research should explore how race/ethnicity and SGM identity modify the associations identified in this study. Finally, there could be other time-varying factors that contribute to the increase in mental health symptoms between our survey waves. Most importantly, the increased media attention to police killings of Black Americans and their daily experiences of discrimination, harassment and microaggressions may have heightened a sense of vulnerability between Waves I and II, particularly for Black college students [45]. While this study cannot speak to the mental health consequences of persistent structural violence towards Black Americans, our results underscore the disparate impacts of Covid-19-related stressors on first-year students' mental health [46, 47].

Colleges across the country have had to make difficult decisions about whether to maintain an in-person semester in the face of the potential concerns of virus transmission or to make classes virtual. As they make these decisions, attention should be paid to college student mental health. Regardless of the choice made, colleges will need to be ready to provide additional counseling support and explore new ways of offering that support virtually [48]. They will need to be creative in providing support for distance learning and helping students to connect safely with each other. They will especially need to thoughtfully engage with Black and SGM students to reduce feelings of social isolation and address sources of structural inequality.

## Supporting information

**S1 Table. Ordinary least squares (OLS) regression estimates for levels of and changes in severity of anxiety and depression symptoms among the Transitions Study longitudinal sample (N = 419).**
(PDF)

## Acknowledgments

We would like to thank two excellent teams of undergraduate researchers at UNC-Chapel Hill who played an integral role in collecting these data, including Michael Almaguer, Caroline

Carpenter, Benjamin Gorman, Luke Hargraves, Susan Huynh, Gabby Goodman, David Lambert, Emilia Mazzolenis, Sarah Parker, Mollie Pepper, and Brittany Wiafe.

## Author Contributions

**Conceptualization:** Jane Cooley Fruehwirth, Krista M. Perreira.

**Formal analysis:** Siddhartha Biswas.

**Funding acquisition:** Jane Cooley Fruehwirth, Krista M. Perreira.

**Methodology:** Jane Cooley Fruehwirth, Siddhartha Biswas, Krista M. Perreira.

**Project administration:** Jane Cooley Fruehwirth.

**Supervision:** Jane Cooley Fruehwirth, Krista M. Perreira.

**Writing – original draft:** Jane Cooley Fruehwirth.

**Writing – review & editing:** Siddhartha Biswas, Krista M. Perreira.

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
