## [Decision Letter · Decision Letter 0]

25 Jan 2021

PONE-D-20-40119

The Covid-19 Pandemic and Mental Health of First-Year College Students: Examining the Effect of Covid-19 Stressors Using Longitudinal Data

PLOS ONE

Dear Dr. Fruehwirth,

Thank you for submitting your manuscript to PLOS ONE. After careful consideration, we feel that it has merit but does not fully meet PLOS ONE’s publication criteria as it currently stands. Therefore, we invite you to submit a revised version of the manuscript that addresses the points raised during the review process.

One reviewer has commented minor revision for your contribution and I agree with his decision. Therefore, I would like to invite you to revise your work after considering his comments. Moreover, I would appreciate that if you can consider the following references on your revision:

Pramukti, I., Strong, C., Sitthimongkol, Y., Setiawan, A., Pandin M. G. R., Yen, C.-F., Lin, C.-Y., Griffiths, M. D., Ko, N.-Y. (2020). Anxiety and suicidal thoughts during the COVID-19 pandemic: A cross-country comparison among Indonesian, Taiwanese, and Thai university students. Journal of Medical Internet Research, 22(12), e24487.

Nathiya D, Singh P, Suman S, Raj P, Tomar BS. Mental health problems and impact on youth minds during the COVID-19 outbreak: Cross-sectional (RED-COVID) survey. Soc Health Behav 2020;3:83-8

Akdeniz G, Kavakci M, Gozugok M, Yalcinkaya S, Kucukay A, Sahutogullari B. A survey of attitudes, anxiety Status, and protective behaviors of the university students during the COVID-19 outbreak in Turkey. Front Psychiatry 2020;11:695

We look forward to receiving your revised manuscript.

Kind regards,

Chung-Ying Lin

Academic Editor

PLOS ONE

Journal Requirements:

2.We note that you have indicated that data from this study are available upon request. PLOS only allows data to be available upon request if there are legal or ethical restrictions on sharing data publicly. For more information on unacceptable data access restrictions, please see http://journals.plos.org/plosone/s/data-availability#loc-unacceptable-data-access-restrictions.

Reviewers' comments:

Reviewer's Responses to Questions

**Comments to the Author**

1. Is the manuscript technically sound, and do the data support the conclusions?

Reviewer #1: Yes

2. Has the statistical analysis been performed appropriately and rigorously? 

Reviewer #1: I Don't Know

3. Have the authors made all data underlying the findings in their manuscript fully available?

Reviewer #1: Yes

4. Is the manuscript presented in an intelligible fashion and written in standard English?

Reviewer #1: Yes

5. Review Comments to the Author

Reviewer #1: Overall, this article showed an interesting topic as it is important to see the changes anxiety level among the students pre and post-pandemic. However, there were several parts need to be clarified as follows:

1. In the background section, the authors mentioned that the first-year students is critical for academic success. What does it mean? What makes it critical?

2. In table 1, the authors listed the demographic characteristic among the two samples (cross-sectional and longitudinal). Why the authors did not include the family income as it mentioned earlier in the background as the related factors.

3. Table 5 looks not clear. Why did the author provide three marginal effect with different values? Why did the sample in column 3,4,7,8 are different? What makes the difference? How did the authors deal with this issue?

4. Still in table 5, why the authors were not able to calculate the odds ratio as this is important to find the likelihood to have high anxiety?

5. On page 18-19, the authors mentioned we found that students who were already experiencing mental health problems pre-pandemic were at greater odds of experiencing severe symptoms mid-pandemic.

Minor comments:

Period should be placed after the citation. Please see the guideline.

6. PLOS authors have the option to publish the peer review history of their article (what does this mean?). If published, this will include your full peer review and any attached files.

Reviewer #1: **Yes: **Iqbal Pramukti, Ph.D.

---

## [Author Response · Author response to Decision Letter 0]

10 Feb 2021

Dear Editor,

Thank you for the opportunity to submit a revised version of The Covid-19 Pandemic and Mental Health of First-Year College Students: Examining the Effect of Covid-19 Stressors Using Longitudinal Data” for your consider for publication as a research article in PLOS ONE. 

We include below our responses to each of your comments and to those of the referee. We put our responses in italics.

We have incorporated the references you mentioned in the text on lines 150-162. They are listed here with their reference number:

(26) Pramukti, I., Strong, C., Sitthimongkol, Y., Setiawan, A., Pandin M. G. R., Yen, C.-F., Lin, C.-Y., Griffiths, M. D., Ko, N.-Y. (2020). Anxiety and suicidal thoughts during the COVID-19 pandemic: A cross-country comparison among Indonesian, Taiwanese, and Thai university students. Journal of Medical Internet Research, 22(12), e24487.

(30) Nathiya D, Singh P, Suman S, Raj P, Tomar BS. Mental health problems and impact on youth minds during the COVID-19 outbreak: Cross-sectional (RED-COVID) survey. Soc Health Behav 2020;3:83-8

(29) Akdeniz G, Kavakci M, Gozugok M, Yalcinkaya S, Kucukay A, Sahutogullari B. A survey of attitudes, anxiety Status, and protective behaviors of the university students during the COVID-19 outbreak in Turkey. Front Psychiatry 2020;11:695

We have addressed the style requirements as requested in your letter. In formatting the tables to meet the style requirement, we divided Table 5 from our previous submission, into two tables (now Tables 5 and 6) to better fit in the document.

We also include the following prompts and our responses with data availability:

The UNC ethics committee has deemed that the data contain potentially sensitive information and that there is a possibility of deductive disclosure, so that the human subjects approval does not allow us to share the de-identified data. The de-identified data will be made available upon request to the Deputy Director of Research at the UNC Carolina Population Center (TransitionsDataRequest@office.unc.edu) with an appropriate restricted use data agreement in place.

We have also added the supporting information captions at the end of the text and updated our Supporting Information files and in-text citations in accordance with the journal guidelines.

Please let us know if there are any other questions or concerns we can address.

Sincerely,

Jane Fruehwirth, Siddartha Biswas and Krista Perreira

Referee Comments with Responses in Italics

Thank you for taking the time to review our article and for the improvements you suggest.

Overall, this article showed an interesting topic as it is important to see the changes anxiety level among the students pre and post-pandemic. However, there were several parts need to be clarified as follows:

1. In the background section, the authors mentioned that the first-year students is critical for academic success. What does it mean? What makes it critical?

Thank you for this comment. We adjusted this text, which is on lines 110-123 to clarify. It now indicates “ This study focuses on a diverse sample of first-year students. The first year is understood to be a particularly challenging year for students given the transition to a new school environment and the increased independence students experience [12,13].” We also added citations to support the statement.

2. In table 1, the authors listed the demographic characteristic among the two samples (cross-sectional and longitudinal). Why the authors did not include the family income as it mentioned earlier in the background as the related factors.

We clarify on line 271 that free/reduced price lunch status is the proxy we have for low-income. This is the measure that is included in Table 1.

3. Table 5 looks not clear. Why did the author provide three marginal effect with different values? Why did the sample in column 3,4,7,8 are different? What makes the difference? How did the authors deal with this issue?

Thanks for pointing this out. Column 3(4) were marginal effects from a logistic regression on students without (with) moderate-severe anxiety symptoms in Wave I. Column 7(8) were marginal effects from a logistic regression on students without (with) moderate-severe depression symptoms in Wave I. We expected results to be different on these subsamples and discuss in the text how results differ for those with and without symptoms in Wave I. Please note that for formatting purposes these results are now split between Table 5 (first 4 columns from previous table with moderate-severe anxiety symptoms as the dependent variable) and Table 6 (last 4 columns from previous table with moderate-severe depression symptoms as the dependent variable), but reported marginal effects are the same.

We explain this more clearly now on lines 400 to 402. This reads:

“The remaining regressions in Table 5 (6) were estimated separately for those with and those without Wave I anxiety (depression) symptoms and did not control for Wave I anxiety (depression) symptoms as a result.”

We also add a row at the bottom of Tables 5 and 6 to indicate that the first two regressions were estimated on the whole sample. We also clarify the other row in the table that indicates whether the sample was estimated on the sample with or without Wave I symptoms. We changed the text from: “Moderate to severe symptoms in W1” to read now “Moderate to severe anxiety symptoms in Wave I?” and a similar row for Table 6, but replacing “anxiety” with “depression”.

4. Still in table 5, why the authors were not able to calculate the odds ratio as this is important to find the likelihood to have high anxiety?

We changed this so that we now report odds ratios for the regressions on the overall sample along with the marginal effects. Previously we just mentioned a preference for marginal effects given that it allows us to compare across model specifications. 

We explain the logic now for the marginal effects in more detail on lines 402 to 413. It reads:

“Odds ratios cannot be compared across model specifications when the sample or conditioning set changes, whereas marginal effects can be compared [43,44].”

This is particularly important in our setting given that we want to compare how effect sizes change across different subsamples of our data and when we change the conditioning set.

5. On page 18-19, the authors mentioned we found that students who were already experiencing mental health problems pre-pandemic were at greater odds of experiencing severe symptoms mid-pandemic. 

This is correct. In our models, we include a covariate for the students’ mental health pre-pandemic at Wave I. We now report both marginal effects (column 2) and odds ratios (column 1). The marginal effect of Wave I anxiety symptoms is 0.16 (Table 5) and the marginal effect of Wave I depression symptoms is 0.32 (Table 6). The corresponding odds ratios are 3.58 and 5.24 respectively. 

Minor comments:

Period should be placed after the citation. Please see the guideline.

These are now fixed. Thanks for pointing out this error.

---

## [Editor Report · Decision Letter 1]

18 Feb 2021

The Covid-19 Pandemic and Mental Health of First-Year College Students: Examining the Effect of Covid-19 Stressors Using Longitudinal Data

PONE-D-20-40119R1

Dear Dr. Fruehwirth,

We’re pleased to inform you that your manuscript has been judged scientifically suitable for publication and will be formally accepted for publication once it meets all outstanding technical requirements.

Kind regards,

Chung-Ying Lin

Academic Editor

PLOS ONE
---

## [Editor Report · Acceptance letter]

22 Feb 2021

PONE-D-20-40119R1 

The Covid-19 pandemic and mental health of first-year college students: Examining the effect of Covid-19 stressors using longitudinal data 

Dear Dr. Fruehwirth:

I'm pleased to inform you that your manuscript has been deemed suitable for publication in PLOS ONE. Congratulations! Your manuscript is now with our production department. 

Kind regards, 

on behalf of

Dr. Chung-Ying Lin 

Academic Editor

PLOS ONE